# Determinants of sustainable solid waste management in Jimma City, Southwest Ethiopia

**Gutama Haile Degefa**[1]*, **Kasahun Eba**[1], **Habtamu Roba**[2], **Mohammedgezali Ibrahim**[1], **Zewdie Birhanu**[3,4], **Temima Jemal**[1], **Worku Jimma**[5], **Fikadu Mitiku**[6,7], **Gudina Terefe Tucho**[1]

**1** Department of Environmental Health Science and Technology, Jimma University, Jimma, Ethiopia, **2** Department of Environmental Health Science, Institute of Health Science, Bule Hora University, Bule Hora, Ethiopia, **3** Department of Health Behavior and Society, Jimma University, Jimma, Ethiopia, **4** JBI Ethiopian Evidence Synthesis, translations and implementation center, Jimma University, Ethiopia, **5** Department of Information Science, Jimma University, Jimma, Ethiopia, **6** Department of Agricultural Economics and Agribusiness Management, Jimma University, Jimma, Ethiopia, **7** Department of Agricultural Economics, Arsi University, Asella, Ethiopia

* gutamahaile@gmail.com

## Abstract

### Background

Exponential urban growth has led to a significant increase in solid waste production, making solid waste one of the most significant issues faced by urban spaces in developing countries. This rising volume of solid waste has led to pressing public health and environmental concerns, such as water, soil, and air pollution, increased greenhouse gas emissions, and the spread of diseases. Thus, this study aimed to evaluate the sustainable solid waste management practices and challenges in Jimma City, southwestern Ethiopia.

### Methods

A community-based cross-sectional study design was employed in this study. Quantitative data and solid waste samples were collected between 01/01/2024 and 01/03/2024 via stratified random sampling from 820 participants in Jimma City, Southwest Ethiopia. The data was analyzed using STATA 18, and a p-value <0.05 was used to determine the level of statistical significance.

### Results

This study revealed a solid waste generation rate of 0.66 Kg/capita/day and the majority of households (84.63%) do not segregate their solid waste at a point of generation; only 38.66% of Households had access to door-to-door solid waste collection services even though about 81.71% of households are willing to pay for solid waste collection services and 69.76% of Households dump waste along rivers or roadsides. Household income, geographic location, level of education, and attitude are the major

**Data availability statement:** All relevant data are within the paper and its Supporting Information files.

**Funding:** The author(s) received no specific funding for this work.

**Competing interests:** The authors have declared that no competing interests exist.

determinants of sustainable solid waste management, with Average Marginal Effects of (0.0411, 0.1098, 0.0621, 0.0495), respectively.

## Conclusion

There is a higher rate of solid waste generation and a lack of integrated solid waste management services like door-to-door collection, temporary public solid waste collection containers, and disposal systems. This study indicated that about 2/3 of total solid waste generation is attributed to organic waste, and limited waste-to-resource recovery practices are observed. Thus, systematic provisions of integrated solid waste management services, implementation of solid waste reduction, and waste-to-resource recovery strategies focusing on composting are recommended.

---

## Introduction

Solid waste refers to any objects or substances that are solid in the state of matter and are unwanted by the producer or owner [1]. Global solid waste generation is increasing rapidly with the rapid increase in population, coupled with high demand for economic growth, technological advancements, and urbanization, placing enormous strain on waste management systems [2]. A solid waste management system is an approach and a process incorporating the physical elements (generation, processing, collection, and treatment), and governance aspects, including the strategies and regulations [3]. Global solid waste generation is expected to increase to 2,590 million tons per year (7.10 million tons/day) by 2030 and 3,400 million tons (9.32 million tons/day) by 2050 [4]. The main contributing factor to the elevation of solid waste generation is believed to be population growth, urbanization, economic development, income levels, and changing consumption patterns [4]. By 2050, about 68% of the global population and 56% of Africa's population will reside in urban areas [5]. This unprecedented pace of urbanization presents unique challenges and characteristics specific to the continent [6,7]. Given the increasing generation rate of solid waste in Africa (81 million tons in 2012–174 million tons in 2016, with projections of 269 million tons by 2030) due to exponential urban growth and industrialization [4,6]. Solid waste management has become one of the most significant issues faced by urban areas in developing countries due to the complexity of the waste composition, lack of infrastructure, economic and institutional alignments, and community attitudes and practices [7,8].

Cities in developing countries generate tons of municipal solid waste daily. In low-income countries, only about 48% of waste is collected in urban areas, while collection coverage drops to approximately 26% in rural areas, reflecting major service delivery gaps [4]. The remaining waste was often left uncollected and ended up in open fields, burned by residents in their backyards, and dumped into waterways, which resulted in significant pollution problems and risks to human health and

the environment [9]. This issue is particularly acute in developing countries, where high population density and economic instability complicate waste sorting and handling [10].

Mismanagement of waste is linked to numerous problems, including water, soil, and air pollution, increased greenhouse gas emissions, and the spread of diseases like urban zoonosis, where rodents and canines serve as reservoirs for disease [11]. The accumulation of solid waste garbage in the environment due to solid waste mismanagement creates burrowing sites and breeding grounds for insects, flies, and mosquitoes, leading to stress, psychological distress, respiratory diseases, diarrheal diseases, malaria, and typhoid fever [12]. On the other hand, solid waste mismanagement leads to unpleasant surroundings, land degradation, depleting flora and fauna, greenhouse gas emissions, obnoxious odor production, and fire hazards [13]. Inappropriate solid waste management leads to significant environmental pollution and health effects, including soil contamination, air pollution, ecological degradation, water pollution, respiratory diseases, and diarrheal diseases [14]. Ethiopia's municipal solid waste generation rate was 0.38 kg/person/day in the year 2024 [15], which is lower than the regional solid waste generation rate, with sub-Saharan Africa generating an average of 0.46 kg/person/day in the year 2016 [4].

In Ethiopia, current solid waste management practices, combined with the country's substantial waste production and uneven waste management, have made solid waste management significantly challenging, affecting human health, environmental quality, and aquatic ecosystems, which can potentially lead to water pollution and increased health risks [16]. This calls on the urgent need for sustainable solid waste management (SSWM) strategies based on real-time data, considering the contributing factors and challenges [16,17]. Most cities and towns in the country reported poor management of solid waste, ineffective waste segregation, collection, transportation, and disposal systems, with the gaps identified as institutional structures, inadequate disposal sites, limited public awareness, and poor infrastructure highlighted as contributing factors [18,19].

Jimma City is one of the largest cities in Ethiopia, with a population of approximately 283,233. Strategically located in the southwest, it serves as an economic hub and a vital link between the Southwest and Gambella Regional States, and Addis Ababa, the capital city of Ethiopia. The city is surrounded by districts with year-round agricultural activity and receives large volumes of fruits, vegetables, khat, and coffee. The import of these agricultural products significantly contributes to the generation of solid waste within the city. In Jimma City, the solid waste generation rate is 0.66 kg/person/day, and the solid waste is composed of 68.34% biodegradable and 29.30% non-biodegradable solid waste. Compared to cities in the Oromia region, Jimma City's solid waste generation rate was relatively higher than that of other cities in the region.

In Jimma city, only 12.32% of the total land area has public solid waste collection containers, and the remaining portion of the city's residents lack adequate waste collection mechanisms, which opens doors to improper solid waste disposal [20]. The solid waste collection system in the city comprises 25% of households using municipal solid waste containers serviced by city trucks, and 2% relying on licensed private micro-enterprises for door-to-door collection. In comparison, the majority 51% practiced open dumping and 22% burned their waste, indicating that 73% of waste disposal occurred through informal and unregulated means [20]. On the other hand, despite the majority of residents (83.5%) expressing willingness to pay for improved solid waste collection services, satisfaction with municipal solid waste management services remains low (25.3% satisfied) [21]. Studies done in Jimma City revealed a lack of comprehensiveness in solid waste management practices, addressing the generation, collection, composition, and disposal system, while some of the studies counted the human and environmental impacts of unregulated solid waste management in the city [22]. Open dumping and uncontrolled burning of household and biomedical waste in Jimma release harmful pollutants, contributing to increased respiratory illnesses [23]. The accumulation of solid waste clogs drainage systems, resulting in stagnant water that promotes mosquito breeding and vector-borne diseases; informal waste collection exposes handlers to hazardous materials, with over 60% of waste workers reporting frequent health issues due to direct contact with medical and household waste [24].

The new Jimma city's urban plans have goals related to transport, drainage, and public spaces, but lack concrete targets, budgets, or performance indicators tied to solid waste management. Even though urban infrastructure development is advancing, the city's development plan does not formally integrate SWM objectives. However, the growth speed and expansion of Jimma City, aligned with uneven data about the city's solid waste management, failed to reflect the current status of solid waste management and keep pace with the city's development plan.

The current study aims to support Jimma City's development plans and policy initiatives by addressing key gaps in solid waste management. It assesses current waste generation patterns, solid waste compositions, household-level sustainable practices, and influencing factors. By providing locally relevant empirical data, the research contributes to more effective urban planning, environmental protection, and sustainable economic development in Jimma City, Southwest Ethiopia.

## Materials and methods

### Study design and setting

A community-based cross-sectional study design was employed. A survey was administered from 01/01/2024–01/03/2024 on selected households by a systematic random sampling technique. Jimma City is 356 kilometers from Addis Ababa, the capital city of Ethiopia. Geographically, the city is located at 7°40′24.47″N latitude and 36°5′4.95″ E longitude [25]. The city has 17 kebeles (the lowest local administrative unit next to district) and an estimated 56,607 households and 283,233 people based on the Jimma Plan and development office data [26].

### The source and study population

The source population comprised all households in Jimma City, representing a diverse urban demographic. The study population consisted of randomly selected households across different neighborhoods to ensure inclusiveness and diversity. The respondents were the household heads. In the absence of a household head, respondents were carefully selected from available and eligible family members aged 18 years or older who were expected to provide full responses to the study.

### Sample size determination and sampling technique

The sample size for the study was determined using a single population proportion formula by Cochran 1977 [27]. With the assumption of a 95% confidence interval, a margin of error of 5%, and a population proportion of 55%, as reported by [28]. Considering a 5% non-response rate and a design effect of 2. A total of 820 respondents participated in the survey.

$$n = \frac{(Z\alpha/2)^2 P(1-P)}{d^2}$$

A stratified sampling technique was employed in which, at the first stage, 17 kebeles (the lower administrative units) were divided into three strata central, middle, and outer kebeles, by considering factors such as proximity to the central business district and municipal administration, population density, waste generation rates, infrastructure and service availability, and other socio-economic attributes on SSWM practices. In the second stage, two kebeles were randomly selected from the 3 strata (Core: Bacho Bore and Hermata Mentina, Middle: Ginjo Guduru and Mendera Kochi, Outer: Jiren and Bore), ensuring representation from each stratum. Finally, six kebeles were selected carefully to balance resource constraints with the need for representativeness, allowing manageable data collection while capturing diverse community contexts within the district. Cluster-adjusted standard errors were considered in the regression analysis to account for the hierarchical structure of the data, where households were nested within kebeles. Finally, households within the selected kebeles were chosen using a systematic random sampling technique, where every nth household was selected after determining a random starting point.

## Data collection tools and procedures

A structured questionnaire and checklist were used to collect the household survey and solid waste composition data. The data collection instruments for socio-economic variables were structured questionnaires adapted from the Ethiopian Demographic and Health Survey, covering key indicators such as residence, sex, age, education, occupation, housing ownership, and income [29]. Sections on attitudes and solid waste management practices were drawn from established KAP tools used in urban studies conducted in Ethiopian towns [30,31]. The questionnaire was prepared in English, translated into local languages (Afaan Oromoo), and pre-tested in a similar kebele in non-study sites before data collection. In the fieldwork, trained data collectors (6 MSc environmental health professionals) collected the data using Kobo tools through face-to-face interviews.

## Solid waste generation and composition

The solid waste quantification and composition were conducted on the sampled households from selected kebeles (the lower administrative units), with each household's solid waste taken continuously for an eight-day waste collection period, and the first-day solid waste was excluded since it would be the accumulation of the previous days. Households were provided daily with three labeled plastic bags for sorting: one for biodegradable waste (e.g., food, leaves, wood, agricultural residues), another one for non-biodegradable waste (e.g., plastics, papers, metals, glass), and one for hazardous waste. Two waste collectors, trained and assigned to each kebele, collected and sorted waste once in the morning to facilitate waste dumping to the proper dumping area after weighing and sorting. The collected waste was then transported by persons, automobiles, and carts to designated sorting sites where further classification took place. The sorted waste was weighed separately by component using a calibrated weighing balance, with data recorded daily for each household on the prepared checklist.

## Solid waste generation rate

The weight of solid waste generation in kilograms per capita per day (Kg/c/day) of household solid waste generation (HSWG) was calculated using the segregated fractions and mixed or total waste collected on a given day [2].

$$PCHSWG(\frac{\frac{Kg}{c}}{day}) = \frac{Total\ weight\ of\ HSW\ generated\ within\ 7\ days}{A\ total\ number\ of\ families\ in\ the\ HH\ \times\ number\ of\ the\ day}$$

## Solid waste composition

The weight of all constituent components in the sample was combined to compute the weight of the overall sample. The composition of each constituent was calculated as a percentage [2].

$$\%age\ composition\ of\ solid\ waste\ fraction\ = \frac{Weight\ of\ separated\ waste}{The\ total\ mixed-weight\ sample} \times 100\%$$

## Study variables

The study examines SSWM practices with households (HHs) as the unit of analysis. The dependent variable is operationalized as SSWM practices, based on the responses to 9 questions that define and characterize SSWM practices. The nine questions used in this study were developed based on a thorough review of peer-reviewed literature and expert consultations [32,33], which guided the selection of essential sustainable solid waste management (SSWM) practices such as segregation, storage, container covering, burning, collection, selling, reusing (direct repurposing without processing,

distinct from recycling (melting/remolding)), composting, and dumping. The questionnaire was further refined through the input of a multidisciplinary author team, comprising five PhD holders (two full professors, two associate professors, and one assistant professor) and four master's degree holders specializing in environmental health and related fields. Finally, the household's SSWM practices were classified into three categories of outcome: unsustainable, moderate, and sustainable.

These categories reflect relative waste hierarchy principles, environmental impact, and resource efficiency. Unsustainable practices involve open dumping, uncontrolled incineration, including open burning, and landfilling without segregation. Moderate practices include partial segregation, recycling, or composting, but lack systematic infrastructure. Sustainable practices align with closed-loop systems, such as source segregation, reuse, energy recovery, and community recycling activities, and the details are in S1 File. After aggregating the survey responses on the nine aspects of SWM, scores were then converted to 100% to classify household SWM practices as "unsustainable" for those ≤33.33%, "moderate" for those ranging from 33.33% to 66.67%, and "sustainable" for those ≥66.67% [34,35].

Meanwhile, the selection of independent variables is grounded in the existing literature on solid waste management (SWM). Socio-demographic factors such as gender and education have been widely recognized as influencing household waste management behaviors [36,37]. Income level is a crucial determinant, as higher-income households often generate more waste but have better access to sustainable practices [38]. Dwelling ownership (owner vs. renter) can influence long-term investment in sustainable waste practices [39]. Moreover, residence duration and solid waste generation rate provide important context on the household's embeddedness in the community and the scale of their waste management needs. The attitudes toward solid waste management are also consistently found to be strong predictors of actual behavior [40]. Finally, the geographic location reflects the spatial disparities in service access and environmental awareness [41]. To this end, gender, education, income, dwelling ownership, residence duration, solid waste generation rate, attitudes, and locational attributes are, thus, included as independent variables in our modeling to capture the complex interplay between socio-economic, behavioral, and spatial factors affecting sustainable solid waste management. The list of variables used in the study is shown in **Table 1** below.

## Data processing, modeling, and analysis

The data were first coded and cleaned using Epi Data 4.6.06, and all statistical analyses were conducted using Stata SE 18 software. The outcome variable, i.e., household SSWM practice, is a categorical variable with three ordered ranks: unsustainable, moderate, and sustainable. This ordered nature requires a modeling approach that accounts for rank ordering without assuming equal spacing between categories [42]. Ordinal logistic regression, particularly the proportional odds model, fits this need by modeling the cumulative odds of being at or below a given category, making it more efficient and interpretable than multinomial models that ignore ordering. In the context of solid waste management studies, several researchers have used ordinal logistic regression to model behavioral, socio-economic, and attitudinal predictors of environmental practices [43,44]. These studies confirm that sustainability outcomes, like waste management practices, often occur on an ordinal scale, justifying the use of ordinal models.

Accordingly, an ordinal logistic regression model was employed to estimate the latent variable underlying the ordinal responses by considering the proportional odds ratio, as follows:

$$Y* = \beta_0 + \beta_1 . Gdr + \beta_2 . Educ + \beta_3 . Inc + \beta_4 . Dwl\_Own + \beta_5 . Res\_Dur + \beta_6 . SWGenR + \beta_7 . Attd + \beta_8 . Kebele + \varepsilon$$

In the above equation, $Y*$ represents an estimate of the latent variable underlying the ordinal response for the observed outcome, i.e., sustainable solid waste management (SSWM) practice, based on thresholds (cut-points); $\beta_0$ denotes the constant term, whereas $\beta_1$ through $\beta_8$ are estimated coefficients of predictors, and represent the error term.

Robust and cluster-adjusted standard errors for heteroscedasticity and intra-kebele correlation, ensuring valid inference was performed. Post-regression estimates, i.e., Average Marginal Effects (AMEs), were computed to predict the impact of

**Table 1. Type and measurement of study variables.**

| Study Variables | Notation | Type/Measurement | Code/Scale |
|---|---|---|---|
| Sustainable Solid Waste Mgmt. | SSWM | Ordinal Categorical | 1: Unsustainable<br>2: Moderate<br>3: Sustainable |
| Gender of the Household Head | Gdr | Nominal Categorical | 1: Female<br>2: Male |
| Education | Educ | Ordinal Categorical | 0: No Formal Educ.<br>1: Primary Educ.<br>2: Secondary Educ.<br>3: Tertiary Educ. |
| Income Category* | Inc | Ordinal Categorical | 1: Low Income<br>2: Middle Income<br>3: High Income |
| Dwelling Ownership | Dwl_Own | Nominal Categorical | 1: Owner<br>2: Rented |
| Residence Duration | Res_Dur | Discrete | Years |
| Solid Waste Generation Rate | SWGenR | Continuous | Kg/HH/day |
| Attitude towards SSWM | Attd | Ordinal Categorical | 1: Unfavourable<br>2: Moderate<br>3: Favourable |
| Geographic Location (Kebele) | Kebele | Nominal Categorical | 1: Jiren<br>2: Hirmata Mentina<br>3: Mendera Kochi<br>4: Ginjo Guduru<br>5: Bacho Bore<br>6: Bore |

*Note that households with a monthly income of ≤ 3500 ETB are categorized as "Low Income"; those with income of >3500 and ≤ 7,000 ETB are regarded as "Middle Income"; while those earning >7000 ETB are "High Income."

independent variables on the likelihood of households adopting sustainable solid waste management practices, offering a more accurate and comprehensive understanding of how these factors influence probabilities in multinomial data [45]. The ditails of data analysis model output was filed in S2. Finally, the results of descriptive statistics and regression estimates were summarized and presented in tables, graphs, and figures, aiding in the clarity and dissemination of key findings.

## Operational definitions

"Solid waste" was defined as waste generated in households by residential dwellings. Sustainable solid waste management, in this context, refers to minimizing solid waste through reduction, reuse, recycling, and energy recovery. Households were considered to practice sustainable solid waste management if they reported actions such as segregating waste, storing waste properly, reusing, recovering resources from solid waste, selling, and composting. This was further cross-checked during observation through the presence of sorted waste, separate containers for different waste streams, evidence of reuse and recycling practices, and proper disposal methods that prevent environmental pollution and public health threats.

## Ethical clearance

Ethical clearance was obtained from the Jimma University Institute of Health, Institutional Review Board (IRB). Before data collection, permission was sought from the Jimma City Municipality and local kebele leaders. Written informed consent was obtained from study participants before data collection. Personal safety measures, the use of personal protective

equipment (PPE), and safety measures for hazardous waste handling, such as placing sharps in puncture-resistant containers labeled with biohazard symbols for proper disposal, segregating chemical waste to ensure non-infectious disposal, and providing training on safe handling, waste segregation, and emergency protocols. Emphasis was also placed on personal hygiene, including thorough handwashing, to minimize contamination risks.

## Results

### Socio-demographic characteristics of respondents

In the study, 820 households participated, resulting in a response rate of 98.09%. The majority of household heads are men (72.68%), and significant numbers of them (40.85%) have a university degree, suggesting that the population is relatively educated and that men dominate leadership roles. Three-fifths of the households (59.76%) reside in owned dwellings, indicating an extent of financial stability and residential permanence, while more than half of them (54.88%) fall into the higher-income category. There is a need for focused awareness initiatives to promote greater proactive engagement in sustainable practices, as nearly 2/3 (63.17%) practices moderate SSWM, while just a quarter (26.34%) of the households exhibited a favorable attitude towards SSWM practices, and the details were in **Table 2**.

### Current household sustainable solid waste management practices in Jimma City

Household solid waste segregation is very low, with only 15.37% of residents engaging in the practice. Resource recovery activities such as composting organic waste (11.34%), reusing (21.83%), and selling recyclable materials (29.39%) were practiced by households, as shown in Table 3.

**Table 2. Socio-demographic and spatial characteristics of the sample households.**

| Household Characteristics | Categories | Frequency | Percentage |
|---|---|---|---|
| Gender of Household Head | Female | 224 | 27.32 |
| | Male | 596 | 72.68 |
| Education | No formal education | 184 | 22.44 |
| | Primary Education | 135 | 16.46 |
| | Secondary Education | 166 | 20.24 |
| | Tertiary Qualification | 335 | 40.85 |
| Income | Lower Income | 152 | 18.54 |
| | Middle Income | 218 | 26.59 |
| | Higher Income | 450 | 54.88 |
| Dwelling | Owner | 490 | 59.76 |
| | Rented | 330 | 40.24 |
| Attitude | Unfavorable | 86 | 10.49 |
| | Moderate | 518 | 63.17 |
| | Favorable | 216 | 26.34 |
| Location | Jiren | 51 | 6.22 |
| | Hirmata Mentina | 108 | 13.17 |
| | Mendera Kochi | 113 | 13.78 |
| | Ginjo Guduru | 84 | 10.24 |
| | Bacho Bore | 199 | 24.27 |
| | Bore | 265 | 32.32 |

The study covered households from six kebeles. The proportional distribution was as follows: Jiren (52), Hermata Mentina (108), Mendera Kochi (118), Ginjo Guduru (85), Bacho Bore (206), and Bore (267). The duration of residence in the sample households was in the range of with a Mean of 13.44 and a standard deviation of 13.36.

Low level of solid waste segregation reduces the possibility of effective composting and recycling, highlighting the necessity of increased awareness to promote SSWM practices.

**Solid waste generation rate and composition**

The household solid waste composition from the surveyed households was analyzed, and the details of each component are shown in **Fig 1** below.

The study revealed that the average solid waste generation rate was 0.66 kg per capita per day. This corresponds to a total daily generation of 174,860.74 kg and an annual accumulation of 63,867,885 kg of municipal solid waste. Composition analysis showed that biodegradable materials accounted for 68.34%, and non-biodegradable materials accounted for 29.30%.

**Door-to-door solid waste collection service, willingness to pay, and household satisfaction level on solid waste management services**

The households' solid waste management services satisfaction level is shown in **Fig 2** below.

**Table 3. Household sustainable solid waste management practices.**

| SSWM practice by type of engagement | Frequency | Percent |
|---|---|---|
| Segregation | 126 | 15.37 |
| Storage | 653 | 79.63 |
| Cover container | 321 | 38.40 |
| Open fields burning | 472 | 57.56 |
| Selling | 241 | 29.39 |
| Reusing | 179 | 21.83 |
| Composting | 93 | 11.34 |
| Dumping along the river and roadside | 572 | 69.76 |

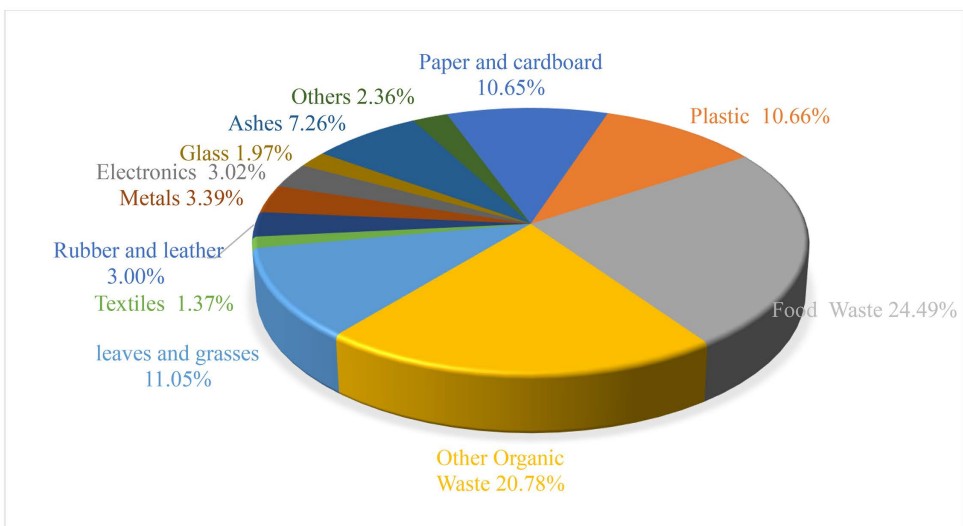

**Fig 1. Composition of household solid waste.**

Based on the survey response of the households, only 38.66% of Households had access to door-to-door solid waste collection services, even though about 81.71% of households are willing to pay for door-to-door solid waste collection services. Thus, the majority of the households were highly dissatisfied with solid waste management services, while only 31.34% were satisfied. The finding highlights the need for better service quality, infrastructure, public education, and reliability to meet community expectations and sustainability goals.

## Results of descriptive statistics

The finding reveals that the gender of the household head, education, income, dwelling ownership, attitude, geographic location, residency duration, and solid waste generation were the predictors of SSWM practices, and the details are in **Table 4**.

Education significantly influences SSWM behavior, with households led by individuals with secondary (33.78%) and tertiary education (32.43%) showing higher sustainable practices compared to those with no formal education (13.51%) ($p < 0.001$). Income level is strongly associated with SSWM, as 63.89% of higher-income households engage in sustainable practices, while only 12.5% of lower-income households engage in SSWM ($p < 0.001$). Furthermore, attitude plays a critical role, with 44.59% of households with favorable attitudes practicing sustainability, in contrast to just 5.41% among those with unfavorable attitudes ($p < 0.001$).

## Results of ordinal logistic regression

The household survey identified key factors determining SSWM. The gender of the household head had a statistically significant effect on SSWM practices. Male-headed households have 1.42 times higher odds of practicing a higher SSWM than female-headed households. The probability of moving to a more sustainable practice increases by 2.42% for male-headed households (Coef. = 0.352). Households where heads completed primary education are 2.02 times more likely to practice SSWM than those with no formal education, which increases the probability of sustainability by 4.66% (Coef. = 0.701). The secondary education is associated with SSWM 2.39 times higher odds; with the probability of increases by 6.21% (Coef. = 0.870) to engage in SSWM practices compared to those with no formal education. Higher-income households are 1.77 times more likely to engage in SSWM practices compared to lower-income households.

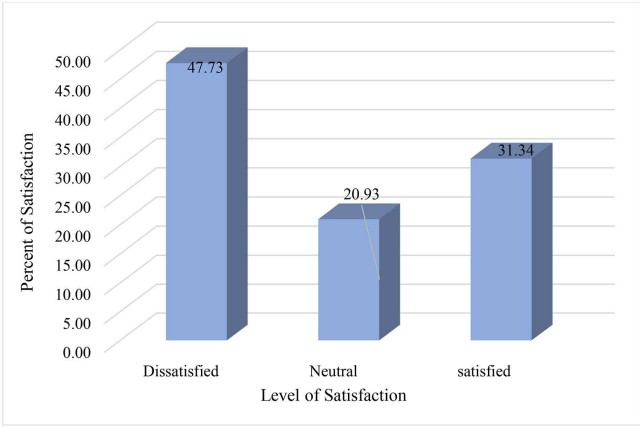

**Fig 2. Household satisfaction level with the solid waste management service.**

**Table 4. Summary of descriptive statistics.**

| Predictors | | Sustainable Solid Waste Mgmt. | | | Descriptive Statistics | | |
|---|---|---|---|---|---|---|---|
| Variables | Covariates | Unsustainable (%) | Moderate (%) | Sustainable (%) | Df. | Chi² | Sig. |
| Gender of Household Head | Female | 34.87 | 24.81 | 24.32 | 2 | 8.7351 | 0.013 |
| | Male | 65.13 | 75.19 | 75.68 | | | |
| Education of HH SWMgr | No formal Educ. | 38.66 | 16.79 | 13.51 | 6 | 70.6528 | <0.001 |
| | Primary Educ. | 18.07 | 16.03 | 20.27 | | | |
| | Secondary Educ. | 18.49 | 18.51 | 33.78 | | | |
| | Tertiary Educ. | 24.79 | 48.66 | 32.43 | | | |
| Income | Lower Income | 25.43 | 16.15 | 12.5 | 4 | 59.5038 | <0.001 |
| | Middle Income | 40.95 | 20.96 | 23.61 | | | |
| | Higher Income | 33.62 | 62.88 | 63.89 | | | |
| Dwelling | Owns | 71.31 | 54.02 | 67.12 | 2 | 21.9553 | <0.001 |
| | Rented | 28.69 | 45.98 | 32.88 | | | |
| Attitude | Unfavourable | 15.13 | 8.97 | 5.41 | 4 | 21.359 | <0.001 |
| | Moderate | 63.45 | 64.31 | 50 | | | |
| | Favourable | 21.43 | 26.72 | 44.59 | | | |
| Location | Jiren | 12.18 | 4.2 | 1.35 | 10 | 71.8085 | <0.001 |
| | Hirmata Mentina | 15.13 | 11.45 | 16.22 | | | |
| | Mendera-Kochi | 13.45 | 13.55 | 20.27 | | | |
| | Ginjo Guduru | 5.46 | 11.45 | 16.22 | | | |
| | Bacho Bore | 35.71 | 20.99 | 14.86 | | | |
| | Bore | 18.07 | 38.36 | 31.08 | | | |

Additionally, households with favorable attitudes towards SSWM had 2.06 times higher odds, which was 4.95% more likely to engage in SSWM practice (Coef. = 0.724) than households with unfavorable attitudes. The geographic location of residence is strongly associated with differences in SSWM practices. Ginjo Guduru shows 7.35 times higher odds with the highest likelihood of 10.98% of SSWM practices compared to those in Jiren (reference kebele) (Coef. = 1.994). On the other hand, Jiren has less likelihood of SSWM practice by 3.68% (Coef. = −1.35) compared to the other kebeles. These variations may stem from localized differences in infrastructure, awareness campaigns, or law enforcement mechanisms. For all of the predictor variables, the 0.05 significance level was considered, and the details are in **Table 5**.

## Discussion

The study focused on solid waste generation, its composition, and the factors influencing sustainable solid waste management practices. According to the household survey study, the current solid waste generation rate was 2.01±2.33 Kg/HH/day (0.41 kg/cap/day), where the average daily solid waste generation rate based on the seven days of solid waste data in the study area was 0.66 Kg/cap/day. The discrepancy between solid waste generation reported in the survey and the seven-day waste quantification data is likely due to self-reporting bias in household waste estimates. According to various studies, the rates of solid waste generation in different towns in Ethiopia vary significantly. For instance, a survey conducted in Dilla town found a generation rate of 0.475 kg/cap/d [46], in Metu town, 0.378±0.05 kg/cap/d was reported [47]. Another study finding in Awaday Town had a reported generation rate of 0.85 kg/cap/d [48]. Compared to the above literature, the present finding was relatively higher than Dilla and Metu and lower than Awaday, i.e., it was in between. This could be attributed to several factors, including differences in the community awareness level of SWM, enforcement of rules and regulations, implementation of SSWM, economic activities, average monthly income, lifestyle, consumption patterns of residents, and sample size.

**Table 5. Ordinal logistic regression output.**

| Dependent Variable: *Sustainable Solid Waste Management* | | | |
|---|---|---|---|
| Summary Statistics | Number of obs. | 820 | |
| | Pseudo R$^2$ | 0.0976 | |
| | Wald Chi$^2$(15) | 112.43 | |
| | Prob. > Chi$^2$ | 0.0000 | |
| **Predictors** | **Coefficients** | **Odds Ratio** | **AME** |
| *Household Head Gender* | | | |
| Male | 0.352** | 1.421 | 0.0242 |
| *Education* | | | |
| Primary Edu. | 0.701*** | 2.015 | 0.0466 |
| Secondary Edu. | 0.870*** | 2.388 | 0.0621 |
| Tertiary & above | 0.415** | 1.514 | 0.0245 |
| *Income Category* | | | |
| Middle Income | −0.268 | 0.765 | −0.0138 |
| Higher Income | 0.568* | 1.765 | 0.0411 |
| *Dwelling Ownership* | | | |
| Rented | 0.092 | 1.097 | 0.0068 |
| *Resid. Duration* | 0.018*** | 1.019 | 0.0014 |
| *Waste Generation/kg* | 0.147*** | 1.158 | 0.0108 |
| *Attitude* | | | |
| Moderate | 0.398 | 1.489 | 0.0238 |
| Favourable | 0.724* | 2.062 | 0.0495 |
| *Kebele* | | | |
| Hirmata Merkato | 1.466*** | 4.334 | 0.0622 |
| Mendera Kochi | 1.443*** | 4.234 | 0.0604 |
| Ginjo Guduru | 1.994*** | 7.345 | 0.1098 |
| Bacho Bore | 0.914* | 2.495 | 0.0293 |
| Bore | 1.733*** | 5.660 | 0.0840 |
| Cutoff 1 | 2.277*** | . | . |
| Cutoff 2 | 6.006*** | . | . |

Legend: * $p<.05$; ** $p<.01$; and *** $p<.001$.

*Source*: Own Computation, March 2025.

The findings indicate that more than two-thirds, 68.34%, of the solid waste generated by households in the study area is biodegradable; 29.30% is non-biodegradable, and 2.36% is other waste, including hazardous waste. This finding is consistent with a previous study conducted in Dilla, which found that a large proportion of generated waste (68.4%) is biodegradable [46]. The high proportion of organic waste shows a strong reliance on agricultural products in the study area.

This study shows that male-headed households were more likely to engage in SSWM practices than female-headed ones. These findings deviate from literature emphasizing women's central role in waste handling; findings across studies are mixed. For instance, a study in Nigeria found male-headed households were more likely to use formal waste management systems than female-headed ones [49]. Another finding conducted in Injibara Town, North East Ethiopia, was consistent with the current finding, where male-headed households practice better solid waste management [50]. On the other hand, a study conducted in southwestern Nigeria observed higher female-headed household participation in solid waste management practices [51]. These differences suggest that gendered waste management behaviors vary by context and

are influenced by factors like education, home ownership, and attitudes [49]. The lower engagement of female-headed households may relate to structural barriers such as time burdens, resource constraints, and cultural roles that limit participation in formal SSWM.

Education was another strong predictor of SSWM, with progressively more effects observed among educational attainment. Secondary education shows a 6.21% increased likelihood of engagement in SSWM practice, and tertiary education shows a 2.45% increased likelihood of engagement in SSWM practices. The findings were consistent with studies previously conducted in Ethiopia, and also align with experiences reported in Bangladesh, where integrated strategies involving education, social norms, and infrastructure have been used to promote circular economy practices and sustainable waste management [52,53]. The high AME in high schools is attributed to secondary education, suggesting enhancement of critical thinking and system understanding. However, the effects of higher education may be undermined by competing social roles and priorities.

Similarly, income levels were highly associated with SSWM practices. Higher-income households showed a 4.11% increased likelihood of engagement in SSWM practices. These findings are similar to household waste segregation being more prevalent in high-income countries [53]. Studies have shown that while high-income households are more likely to engage in SSWM due to their ability to pay for improved services or adopt advanced technologies, lower-income households often rely on informal systems such as open dumping or burning solid waste [4]. This discrepancy might suggest that stable economies enable households to invest in sustainable technologies and practices by purchasing containers and adopting better waste management practices, as they can more easily afford the expenses associated with SSWM.

Residence duration has a marginal positive effect on SSWM, with each additional year of residence associated with a 0.14% more likelihood of engaging in SSWM practices. Longer residence duration enhances familiarity with local waste management systems, encouraging more participation in SSWM practices [54]. Residents with extended stays are more likely to understand the environmental and health risks of poor waste management, motivating behaviors such as segregation and recycling [4,8]. The current finding suggests that longer-term residents develop stronger connections to their communities, leading to greater awareness and commitment to sustainable practices.

Households generating more solid waste volumes show marginal positive SSWM practices, with a 1.08% increased likelihood per additional kilogram of solid waste generated in a day. Findings on urban solid waste management show that households generating more waste tend to engage more in segregation, recycling, and composting [55,56]. In Jimma City, the positive link between higher solid waste generation and sustainable waste management practices indicates that households generating more waste often possess greater economic capacity, can afford solid waste collection services, and are more likely to invest in materials and systems for proper sorting, storage, and disposal. The new corridor development in Jimma City emphasizes the need for urban greenery, which has increased interest in composting practices. There are possible indications that the recent ban on khat markets within the city might be associated with improvements in solid waste management practices; however, this relationship has not yet been formally studied. Before December 2023, khat markets were widespread throughout Jimma, including in collective areas and on main roadsides. Merchants sold khat and often disposed of leftover waste and shielding leaves illegally along roadsides, business areas, and ditches. Since the ban, khat markets have been consolidated into five main market centers at the margin of the city, where merchants are now responsible for transporting their waste to the municipal landfill for proper disposal. This consolidated and organized khat waste transportation might reduce illegal solid waste dumping in the city.

Similar households with favorable attitudes towards SSWM were 4.95% more likely to engage in SSWM practices. This finding is consistent with studies conducted in Ethiopia, where strong attitudes toward waste management are often linked to higher participation rates in sustainable practices such as recycling and composting [15]. These findings suggest attitudes are essential for SSWM adoption, as they encourage engagement and participation in management systems.

The geographic location of residence has a highly significant correlation with SSWM practices, where the highest likelihood, 10.98% of SSWM practices observed in Ginjo Guduru kebele, compared to those in Jiren (reference kebele). On

the other hand, Jiren has a lower likelihood of SSWM practice by 3.68% compared to the other kebeles. Similar patterns were observed in Kebridar City, Ethiopia, where kebeles with better infrastructure had higher waste collection compared to areas with limited access to services [56]. Another study conducted in Addis Ababa, Ethiopia, shows that districts with favorable topographical conditions and better municipal planning exhibited more sustainable solid waste management compared to districts with poor infrastructure [57]. These disparities highlight the role of geographical and infrastructural factors in shaping solid waste management practices. For instance, kebeles like Ginjo Guduru may benefit from improved waste collection systems due to proximity to the central business district and municipal administration. While peripheral or less accessible areas like Jiren face challenges such as infrequent waste collection and reliance on informal disposal methods [4]. The current finding shows that urban centers with higher proximity to administrative centers and better infrastructure often achieve better SSWM practices due to better municipal focus and resource allocation.

Furthermore, the findings on dwelling ownership status (rented) have a marginally positive effect of 0.68% on SSWM practices, better than those owned dwellings, though not statistically significant at any conventional level. The results of previous studies regarding dwelling ownership and SSWM varied. Study findings in Dire Dawa, Ethiopia, suggest that households, regardless of ownership status, may rely on external services for waste management, potentially influencing their engagement in SSWM practices [58]. Findings in Kenya indicate that tenant and landlord households equally struggled with waste segregation and recycling due to systemic issues like inadequate infrastructure and limited awareness [59]. Similar to the current finding, a study in Debre Berhan and Gondar City indicated that the majority of residents lived in condominium-rented houses, practicing proper on-site waste handling practices, suggesting that renters in multi-unit dwellings may have structured waste management systems in place [34,35]. Households in rented dwellings in Jimma City show that SSWM is possibly due to residing in shared compounds, feeling social pressure to maintain cleanliness and avoid conflicts, having limited space to store waste, and being accountable to landlords and neighbors. Owners may also enforce rules, and renters often have easier access to waste services in urban areas.

The solid waste composition analysis revealed that biodegradable waste comprised 68.34%, while non-biodegradable waste accounted for 29.30%. During field observation, we have noted that rubber and leather products, such as shoes, bags, jackets, and belts, are the most common waste in the city. Plastics, mainly polyethylene terephthalate (PET) from water, drink, and cooking oil bottles, and low-density polyethylene (LDPE) from carrier, bread, and food storage bags, constitute the major portion of plastic waste in the city.

Despite this potential, the solid waste has a 68.34% high biodegradable composition. The study findings indicate that only 11.34% of households are currently composting, indicating a substantial discrepancy between the types of waste produced and the management practices applied. Many agree that composting is an economical and sustainable way of managing organic waste. Food scraps, vegetable peels, yard waste, and other biodegradable materials are transformed into nutrient-rich compost that can be used for agriculture and the environment [60,61]. Composting offers advantages to the environment beyond its practicality, such as reducing greenhouse gas emissions, particularly methane from landfills, reducing reliance on artificial fertilizers, and enhancing soil fertility [62]. It also supports resource recovery, climate resilience, and clean energy initiatives in developing countries.

Despite these advantages, the low rate of composting practice (11.34%) could be explained by a lack of institutional support, technical know-how, awareness, and composting infrastructure. Effective composting necessitates both changes in behavior and an environment that supports it, such as incentives, working space, tools, and training [63]. In Jimma City, expanding composting has advantages, including reducing the amount of solid waste dumped into landfills and the nearby environment, reducing pollution levels, and encouraging urban farming and revenue generation. The disparity between the high percentage of biodegradable solid waste and the low composting practice seen could be reduced with the encouragement of community-based composting models and integration/application into municipal solid waste management strategies. Hence, this finding highlights the pressing need to advance composting as a key element of SSWM initiatives in the area.

## Limitations of this study

This study was limited to household solid waste generation, composition, and the factors influencing sustainable solid waste management practices within these domains. Geographically, the research was confined to Jimma City, and thus the findings may not be generalizable to larger urban settings, though they may serve as a reference. Temporally, the study adopted a cross-sectional (latitudinal) design, focusing only on the current state of sustainable solid waste management without capturing seasonal or temporal variations. Future research should incorporate longitudinal data and include a wider range of waste sources to provide a dynamic understanding of sustainable solid waste management.

## Conclusions

The study area exhibits a high rate of solid waste generation but lacks integrated solid waste management (ISWM) services, such as door-to-door collection, public solid waste collection containers, and proper disposal systems. The findings also identify income, educational level, and geographic location as key predictors of sustainable solid waste management (SSWM) practices. Households with higher income and education levels were more likely to engage in SSWM, underscoring the influence of economic resources and awareness provision. Additionally, kebeles located closer to the central business district and municipal administration demonstrated significantly better practices, pointing to geographic disparities. These results suggest the need for targeted, location-specific interventions, community-based initiatives, and improved infrastructure in areas with lower levels of SSWM.

The study revealed that nearly two-thirds of the total solid waste generated in Jimma City is organic, yet practices for recovering waste as a resource remain minimal. To address this, integrated solid waste management systems and effective waste reduction strategies should be prioritized. Given the dominance of biodegradable waste, expanding composting initiatives could greatly reduce dependence on landfills. Furthermore, the long-term sustainability of SSWM can be strengthened through targeted socioeconomic and infrastructure-based interventions.

## Recommendations

Based on the findings of the study, the following recommendations were forwarded.

To improve sustainable solid waste management (SSWM) in Jimma City, the municipality should provide public education/ awareness for the community on sustainable waste management practices, particularly among low-educated households. The city should introduce incentive schemes, such as waste-for-cash initiatives, financial support, or micro-loans could also be provided to low-income households and groups engaged in composting or recycling to encourage their participation and scale their impact. Expanding waste collection coverage and strengthening kebele and sub-kebele waste management infrastructure, especially in underserved peripheral kebeles, is recommended. Market linkage between compost producers and consumers through the facilitation of contracts between composting groups and local farmers or urban gardening cooperatives.

## Supporting information

**S1 File. SWM sustainability level practices classification.**
(DOCX)

**S2 File. Data analysis model outpout.**
(DOCX)

## Acknowledgments

We acknowledge Jimma University, Institute of Health, for supporting this study. We gratefully acknowledge the study participants for their active and voluntary participation. We thank the municipality workers and kebele administrators for their support during data collection.

# Author contributions

**Conceptualization:** Gutama Haile Haile, Kasahun Eba, Gudina Terefe Tucho.

**Data curation:** Gutama Haile Haile, Kasahun Eba, Habtamu Roba, Gudina Terefe Tucho.

**Formal analysis:** Gutama Haile Haile, Kasahun Eba, Gudina Terefe Tucho.

**Funding acquisition:** Gutama Haile Haile, Kasahun Eba, Mohammedgezali Ibrahim, Zewdie Birhanu, Temima Jemal, Worku Jimma, Fikadu Mitiku, Gudina Terefe Tucho.

**Investigation:** Gutama Haile Haile, Kasahun Eba, Gudina Terefe Tucho.

**Methodology:** Gutama Haile Haile, Kasahun Eba, Habtamu Roba, Zewdie Birhanu, Temima Jemal, Worku Jimma, Fikadu Mitiku, Gudina Terefe Tucho.

**Project administration:** Gutama Haile Haile, Kasahun Eba, Mohammedgezali Ibrahim, Gudina Terefe Tucho.

**Resources:** Gutama Haile Haile, Kasahun Eba, Mohammedgezali Ibrahim, Zewdie Birhanu, Temima Jemal, Worku Jimma, Fikadu Mitiku, Gudina Terefe Tucho.

**Software:** Gutama Haile Haile, Habtamu Roba.

**Supervision:** Gutama Haile Haile, Kasahun Eba, Habtamu Roba, Mohammedgezali Ibrahim, Temima Jemal, Worku Jimma, Fikadu Mitiku, Gudina Terefe Tucho.

**Validation:** Gutama Haile Haile, Kasahun Eba, Gudina Terefe Tucho.

**Visualization:** Gutama Haile Haile, Kasahun Eba, Gudina Terefe Tucho.

**Writing – original draft:** Gutama Haile Haile.

**Writing – review & editing:** Gutama Haile Haile, Kasahun Eba, Mohammedgezali Ibrahim, Zewdie Birhanu, Gudina Terefe Tucho.

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
