## [Decision Letter · Decision Letter 0]

3 Jun 2025

PONE-D-25-22742Determinants of sustainable solid waste management in Jimma City, Southwest EthiopiaPLOS ONE

Dear Dr. Haile,

Thank you for submitting your manuscript to PLOS ONE. After careful consideration, we feel that it has merit but does not fully meet PLOS ONE’s publication criteria as it currently stands. Therefore, we invite you to submit a revised version of the manuscript that addresses the points raised during the review process.

 Both reviewers are largely complimentary of the paper but both propose some minor changes to improve it.

We look forward to receiving your revised manuscript.

Kind regards,

Alison Parker

Academic Editor

PLOS ONE

Journal Requirements:

3. In the online submission form, you indicated that all relevant data are within the manuscript and its Supporting Information files. If additional data required including raw data of the survey, it will be provided upon request.

5. Please remove all personal information, ensure that the data shared are in accordance with participant consent, and re-upload a fully anonymized data set.

Reviewers' comments:

Reviewer's Responses to Questions

**Comments to the Author**

1. Is the manuscript technically sound, and do the data support the conclusions?

Reviewer #1: Partly

Reviewer #2: Yes

2. Has the statistical analysis been performed appropriately and rigorously? 

Reviewer #1: Yes

Reviewer #2: Yes

3. Have the authors made all data underlying the findings in their manuscript fully available?

Reviewer #1: No

Reviewer #2: Yes

4. Is the manuscript presented in an intelligible fashion and written in standard English?

Reviewer #1: Yes

Reviewer #2: Yes

5. Review Comments to the Author

Reviewer #1: This research will make an important contribution to the currently insufficient data on waste arisings in non-OECD countries and will be a useful resource for local and regional policymakers. I am excited to see the final version!

However, there are some minor changes and improvements that are needed in order to increase the quality and impact of this research. I am presenting these as a series of questions, which I hope you can answer, followed by some comments:

1) Are waste volumes increasing only because of urban growth, or because of population growth and changes in consumption?

2) Is there any data to support the claim on line 88 that the country has 'enormous' waste arisings? Are waste arisings they larger than in neighbouring countries?

3) On line 98, how does the quantity of waste being generated in Jimma City compare to elsewhere? Are the figures given higher or lower than elsewhere. Figures from neighbouring locations given later in the document could be moved here for context.

4) On line 100, who is emptying these containers? Is there information about how formal or informal collections are undertaken that you can share?

5) On line 105 - what are the actual discrepancies? The descriptive words used are too vague. Do you mean 'differences' rather than discrepancies?

6) Lines 105/106 refers to data on the human and environmental effects of waste management in the city - this sounds really interesting. Can you add some lines to say what they are? This adds to the importance of this research because you are suggesting ways to avoid future negative effects of poor waste management.

7) In Line 109 - does the development plan contain any waste management objectives that you can address?

8) In line 140, by 'far' do you mean 'outer'. Far is an unusual word to use in this context.

9) In line 152 - was any guidance used to help design the questionnaire? Was it based on a similar study? Can a link be provided to it?

10) In line 183, by 'reuse' do you mean recycling? It is different to reuse which may happen in the home or commercially.

11) In line 185, does uncontrolled incineration encompass open burning?

12) In lines 353-356: Are these locations comparable to the study location? If these locations are completely different, then the comparison to them needs to state this. Are the towns with higher waste arisings bigger cities, and the ones with less smaller?

Line 477: was any record made of whether food waste was cooked or uncooked which could influence whether it could be avoided? Also, was the time of year a factor? Were there more leaves than other times of the year for example (or fewer)?

Also, on line 59, the terms 'hardware' and 'software' are not common terms in this context and the former is not used by the authors quoted. Would suggest the terms 'physical elements' and 'governance aspects' including strategies and regulations.

Line 61: different referencing style used here and elsewhere. Please use the same style throughout.

Line 97: please put Jimma City in context. E.g. it is the xx biggest city in Ethiopia with a population of xxx,000 plus location etc. Global readers may not be familiar with the city.

In the introduction, please state more specific aims of the research - to contribute to the development plan or policies for example? What gaps are being filled?

Line 124: please add ", or local administrative wards," after kebeles. The definition is explained later in the document but this needs to be done the first time the word is used.

Line 141: change 'proximate' to 'proximity'

Lines 187/188: It would be useful to have a table in an appendix that explains the different category descriptions in more technical detail. Many will be quite simple and self-explanatory, but some could be open to misinterpretation if not explained clearly.

Line 265: suggest changing 'their' to 'owned'

Lines 279/80: this sentence hasn't been concluded. Could delete "As a result" and end it with "as shown in Table 3."

Lines 288/289: this graph needs to be clearer. Total composition. Spread out the labels. Include complete legend showing all materials included in the audit. A more detailed breakdown of the waste would be useful. Cooked and uncooked organic waste, different types of plastic and metals etc. Rubber and leather are very different materials - why are they together (shoes?). If the data isn't available then perhaps add observational data. Were the plastics predominantly a certain item for example.

Lines 288/289: look at how graphs are presented in other papers - you don't need the boxes around the text. Satisfation should have a capital S.

Lines 310/11 - Table 4: the figs in the right hand column should be < 0.001 and not 0

Line 329: higher odds? The use of this term here and going forward is confusing. Do you mean higher/increased likelihood or increased probability? Suggest describing what you mean by this and then make reference to the table.

Line 331-3: this sentence isn't very clear - suggest rewording it.

Line 350: compared to the figure for the same city cited in line 98?

Line 410: suggest briefly explaining what these are and why banning them makes a difference

Line 432/433: So renting has a positive impact? Results suggested ownership had this impact. Please clarify.

Reviewer #2: From my reading, this article offers an important contribution to the study of sustainable solid waste management (SSWM) by examining the socio-demographic, attitudinal and spatial factors shaping household practices in Jimma City. The authors use survey data from households alongside waste sampling, using ordinal logistic regression to identify key predictors. The study highlights a particular disconnect, the low adoption of composting and recycling despite biodegradable waste being the most common.

Introduction:

The introduction sets the topic well, situating it within both global and regional contexts and referencing relevant literature to establish its significance. However, there are some minor inconsistencies in reference formatting, for example, in lines 61 and 69 that should be standardised. I also recommend checking the accuracy of some citations: reference 5 (Kaza et al., 2018) is cited in relation to population growth figures in Africa, but these figures do not appear in that source. You may find that reference 4 contains the relevant data. In addition, references 5 and 7 appear to be duplicated in the bibliography. The 30–50% statistic attributed to source 11, I could not locate; it may originate from Kaza et al? but this should be confirmed and clearly referenced.

Methodology:

The methodology is clear and well explained. The use of stratified sampling based on kebele characteristics adds strength by enhancing representativeness. The sample size calculation using Cochran’s formula is appropriate, with assumptions stated clearly. The statistical analysis appears appropriate and is carried out with reasonable rigour. Descriptive statistics are well presented and effectively summarise key variables across different levels of sustainability. One area that would benefit from clarification is the 9-question scoring tool used to assess SSWM practices. Was this adapted from a validated instrument or developed specifically for this study? Further detail on this would improve transparency and support reproducibility. It would also be helpful to briefly explain why the study was limited to six kebeles, whether this was for reasons of representativeness, resource constraints, or other practical considerations.

Results:

The results section is thorough and informative. To improve readability, reducing some repetition between the text and tables would help. There is also an inconsistency regarding sample size: the methods mention 836 households surveyed, but the results section starts of saying 820 households participated. Table 2 mostly sums to 836, so this discrepancy needs clarification and consistent reporting throughout. Minor rounding differences appear too, for instance, “31.37%” satisfaction in the text versus “31.34%” in Figure 3; these should be aligned for accuracy. There is also a typo error in the percentage reported on line 327.

Discussion and Conclusion:

The discussion appropriately places findings within Ethiopian and wider literature, indicative of thoughtful interpretation. However, a few reference issues should be addressed. Reference 41 does not appear to include the cited statistic, though the author has another relevant publication (Quantification and Characterization of MSW) that may be more appropriate. The daily waste generation figure in reference 43 (0.60kg versus 0.56kg) should also be checked. Reference 49 discusses Bangladesh; it would be helpful to note this explicitly to avoid confusion with the Ethiopian context. Formatting issues (line 361) need review. Finally, the claim that the ban on the Khat market has influenced waste practices (line 410) should either be supported with a citation or reframed more cautiously rather than a confirmed finding. The conclusion offers a clear and concise summary of the main findings and their implications. Your recommendations, particularly those related to composting and decentralised waste management are timely and actionable.

Limitations:

This section is well presented and transparent.

Additional points:

No concerns were identified in relation to dual-use research or ethical risks.

Overall assessment:

This manuscript addresses an important topic with well-designed methodology and clear implications for solid waste management policy in Ethiopia. Clarifying inconsistencies noted, verifying references and improving methodological transparency would strengthen the paper. I encourage the authors to revise accordingly as the study shows strong potential for publication in PLOS ONE.

6. PLOS authors have the option to publish the peer review history of their article (what does this mean? ). If published, this will include your full peer review and any attached files.

**Do you want your identity to be public for this peer review?** For information about this choice, including consent withdrawal, please see our Privacy Policy .

Reviewer #1: **Yes: ** Elizabeth Cullen

Reviewer #2: No

---

## [Author Response · Author response to Decision Letter 1]

24 Jul 2025

Authors’ responses to the editor’s and reviewers' questions and comments

We appreciate the editor and the reviewers for their rational and constructive comments and thorough revision of our paper. Your valuable and insightful comments led to possible improvements in the current version. We have addressed all comments, including clarifications on methodology, data presentation, and contextual details about Jimma City. We hope the manuscript, after careful revisions, meets your high standards. The authors welcome further comments, if any. All modifications in the manuscript have been tracked in red color.

Response to editor comments

Comment 1: Please ensure that your manuscript meets PLOS ONE's style requirements, including those for file naming.

Response: Thank you for your reminder. We have carefully reviewed our manuscript and confirm that all writing styles and file naming conventions fully comply with PLOS ONE’s style requirements.

Comment 2: Your ethics statement should only appear in the Methods section of your manuscript.

Response: Thank you for your guidance. We have ensured that the ethics statement appears exclusively in the Methods section of our manuscript as required.

Comment 3: In the online submission form, you indicated that all relevant data are within the manuscript and its Supporting Information files.

Response: Thank you for your note. We confirm that all relevant data are included within the manuscript and its Supporting Information files, and any additional data, including raw survey data, will be made available upon request; this statement has also been included in the manuscript.

Comment 4: We note that Figure 1 in your submission contains [map/satellite] images, which may be copyrighted.

Response: Thank you for your critical comments. To address the copyright concern, we have removed the previously submitted image, and we have described Jimma City’s location as 7°40′24.47″N latitude and 36°5′4.95″E longitude according to Abebe et al., 2019, cited in the revised manuscript.

Comment 5: Please remove all personal information, ensure that the data shared is by participant consent, and re-upload a fully anonymized data set.

Response: Thank you for your reminder. We confirm that no personal information has been included in the uploaded data, and we have ensured full compliance with PLOS ONE guidelines regarding participant consent and data anonymization.

Comment 6: Please include captions for your Supporting Information files at the end of your manuscript, and update any in-text citations to match accordingly.

Response: Thank you for your guidance. We have included captions for all Supporting Information files at the end of the manuscript and updated the in-text citations accordingly, per the guidelines.

Comment 7: Please review your reference list to ensure that it is complete and correct.

Response: We have thoroughly reviewed our reference list and confirm that it is complete and accurate, with no retracted papers cited.

REVIEWER 1

Comments to the Author:

General comments: important topic and useful resource for local and regional policymakers!

Response: Thank you very much

Comment 1: Are waste volumes increasing only because of urban growth, or because of population growth and changes in consumption?

Response: Lines 65–66 now clarify that waste volumes are influenced by urban growth, population growth, and changing consumption patterns

Comment 2: Is there any data to support the claim on line 88 that the country has 'enormous' waste arisings? Are waste arisings larger than in neighboring countries?

Response: Thank you. We accepted the comment and edited accordingly.

Comment 3: On line 98, how does the quantity of waste being generated in Jimma City compare to elsewhere? Are the figures given higher or lower than elsewhere? Figures from neighboring locations given later in the document could be moved here for context!

Response: We accepted the comment and edited accordingly.

Comment 4: Line 100, who is emptying these containers? Is there information about how formal or informal collections are undertaken that you can share?

Response: Thank you. We accepted the suggestion and included the solid waste collection system of the town accordingly with citation.

Comment 5: line 105 - What are the actual discrepancies? The descriptive words used are too vague. Do you mean 'differences' rather than discrepancies?

Response: Thank you very much for the clarity questions. We have accepted the comment and replaced the word discrepancy with lack of comprehensiveness. We have used the word discrepancy to show a lack of comprehensive study on solid waste management, and most of the study conducted in the city was focused on one or two of the solid waste management components and hierarchy.

Comment 6: Lines 105/106 refer to data on the human and environmental effects of waste management in the city - this sounds really interesting. Can you add some lines to say what they are? This adds to the importance of this research because you are suggesting ways to avoid future negative effects of poor waste management.!

Response: We have accepted the comment and added sentences on the health and environmental effects of solid waste with citations within the text.

Comment 7: Line 109 - Does the development plan contain any waste management objectives that you can address?

Response: Thank you for your critical questions. Dear reviewer, there is no clear plan for the solid waste management in the new developmental plan. The new developmental plan known as corridor development solely focuses on the drainage system, walkways, recreation, and urban beauty.

Comment 8: line 140, by 'far' do you mean 'outer'. Far is an unusual word to use in this context.

Response: Thank you, reviewer, for your suggestions. We have agreed on the suggested words and replaced the word far with outer.

Comment 9: line 152 - was any guidance used to help design the questionnaire? Was it based on a similar study? Can a link be provided to it?

Response: Thank you very much. We have used guidance in questionnaire design based on an Ethiopian demographic health survey and a similar study from different sources. We have accepted the comment and included a sentence about the data collection tools, and the references were attached in the revised manuscript.

Comment 10: line 183, by 'reuse' do you mean recycling? It is different from reuse, which may happen in the home or commercially.

Response: Thank you very much for the clarity questions. When we say reuse, we are not saying recycling. We define "reuse" as direct repurposing without processing, distinct from recycling (melting/remolding). The reusing in our context is not different from the one happening at the commercial level. We have inserted the definition in the revised manuscript.

Comment 11: line 185, does uncontrolled incineration encompass open burning?

Response: Thank you very much for the clarity comments. Yes, open burning is a subset of uncontrolled incineration, as noted in the revised text.

Comment 12: lines 353-356: Are these locations comparable to the study location? If these locations are completely different, then the comparison of them needs to state this. Are the towns with higher waste arisings bigger cities, and the ones with less smaller?

Response: Thank you for your valuable comments. We agree with your observations and have revised the comparison accordingly. Upon rechecking, we confirmed that the selected Ethiopian towns used for comparison (Dilla, Metu, and Awaday) are all zonal-level towns, which are contextually comparable to Jimma City. We have removed the data from Kano State, Nigeria, as it represents a state-level or regional-scale entity, which is significantly larger than zonal towns in the Ethiopian context and not directly comparable. Regarding waste generation, we observed that the cities we have used to compare exhibit a waste generation rate that is generally consistent with their population size. However, Awaday Town is an exception due to its significant commercial activity, particularly the large khat market, which contributes to a higher per capita waste generation rate compared to other towns, including Jimma.

Comments: Line 477: Was any record made of whether food waste was cooked or uncooked, which could influence whether it could be avoided? Also, was the time of year a factor? Were there more leaves than at other times of the year for example (or fewer)?

Response: Thank you for the insightful comment. In our study, food waste was collected as a general category without distinguishing between cooked and uncooked items, which we acknowledge as a limitation and will consider in future research. As noted in the limitations section, data collection was conducted during the dry season only. Seasonal variation, such as increased leaf waste from green vegetables, khat, and maize peels during the wet season, may lead to more solid waste generation. We have recommended that future studies include seasonal comparisons for a more comprehensive analysis.

Comments: Line 59, the terms 'hardware' and 'software' are not commonly used in this context, and the authors do not use the former term. Would suggest the terms 'physical elements' and 'governance aspects', including strategies and regulations.

Response: Thank you for the helpful suggestion. We have replaced the terms ‘hardware’ and ‘software’ with ‘physical elements’ and ‘governance aspects’, including strategies and regulations, as suggested.

Comments: Line 61: different referencing style used here and elsewhere. Please use the same style throughout.

Responses: Thank you for pointing this out. The referencing inconsistencies have been corrected, and a uniform citation style has been applied throughout the manuscript.

Comments: Line 97: Please put Jimma City in context. E.g. it is the xx biggest city in Ethiopia with a population of xxx,000 plus location etc. Global readers may not be familiar with the city.

In the introduction, please state more specific aims of the research - to contribute to the development plan or policies for example? What gaps are being filled?

Response: Thank you for your insightful comments. We have incorporated comprehensive contextual information about Jimma City and articulated the specific aims of the study. These revisions are included in the introduction section, where the city’s demographic, geographic, and economic significance are described. Additionally, we have detailed the study’s objectives, emphasizing its contribution to the city’s development plans and policies by addressing gaps in solid waste management and urban planning.

Comments: Line 124: please add ", or local administrative wards," after kebeles. The definition is explained later in the document but this needs to be done the first time the word is used.

Responses: Thank you for your suggestion. We have incorporated the phrase "the lowest local administrative unit next to district," immediately after the mention of "kebeles" as requested.

Comments: Line 141: change 'proximate' to 'proximity'

Responses: Thank you for your suggestions. We have made the suggested changes: The word “proximate” has been changed to “proximity” as recommended.

Comments: Lines 187/188: It would be useful to have a table in an appendix that explains the different category descriptions in more technical detail. Many will be quite simple and self-explanatory, but some could be open to misinterpretation if not explained clearly.

Responses: Thank you for your valuable comments. We have accepted the suggestions and inserted the table showing the sustainability categorization variables and the table showing the analysis result of sustainability categorization in supplementary tables 3 and 4.

Comments: Line 265: suggest changing 'their' to 'owned'

Responses: Thank you for your suggestion. We have accepted the recommended change from "their" to "owned" and updated the document accordingly.

Comments: Lines 279/80: This sentence hasn't been concluded. Could delete "As a result" and end it with "as shown in Table 3."

Responses: Thank you for your comment and suggestions. We have revised the sentence to ensure clarity and completeness by removing the phrase “As a result” and properly concluding it. The corrected sentence now reads: “Resource recovery activities such as composting organic waste (11.34%), reusing (21.83%), and selling recyclable materials (29.39%) were practiced by households, as shown in Table 3 in the text.”

Comments: Lines 288/289: this graph needs to be clearer. Total composition. Spread out the labels. Include complete legend showing all materials included in the audit. A more detailed breakdown of the waste would be useful. Cooked and uncooked organic waste, different types of plastic and metals etc. Rubber and leather are very different materials - why are they together (shoes?). If the data isn't available then perhaps add observational data. Were the plastics predominantly a certain item for example.

Responses: Thank you for your detailed feedback regarding the solid waste composition graph. We accept your suggestions and made justifications as follows. We have different organic solid waste data in the analysis presented in the new figures. However, the data did not differentiate between cooked and uncooked food waste, which limits the ability to provide that specific breakdown. Regarding rubber and leather, they were grouped based on the available data, which did not allow for separate categorization. We acknowledge that these are distinct materials, and this limitation is noted in the discussion. Where data were available, other solid waste categories such as leaves and grasses, food waste, and other organic wastes were included in the figures. The hidden parts of the legend were also shown in the revised manuscript. Unfortunately, detailed sub-categorization of plastics (e.g., types or predominant items) was not captured in the solid waste audit data. To address this, we have added observational notes in the manuscript to provide qualitative insights on the nature of rubber and leather, and plastic waste encountered. We appreciate your suggestion and will consider more detailed waste characterization in future studies to enhance clarity and completeness.

Comments: Lines 288/289: look at how graphs are presented in other papers - you don't need the boxes around the text. Satisfaction should have a capital S.

Responses: Thank you for your valuable suggestion. We have removed the boxes around the text and corrected the spelling by capitalizing the "S" in "Satisfaction" as recommended.

Comments: Lines 310/11 - Table 4: the figs in the right-hand column should be < 0.001 and not 0

Responses: Thank you for your careful observation. We agree with your comment and have corrected the p-value presentation in Table 4. The values previously shown as 0.000 have been updated to < 0.001. This change has been made throughout the manuscript wherever applicable.

Comments: Line 329: higher odds? The use of this term here and going forward is confusing. Do you mean higher/increased likelihood or increased probability? Suggest describing what you mean by this, and then refer to the table.

Responses: Thank you for your insightful comment. We have clarified the terminology by replacing "higher odds" with "more likely" and provided a clear explanation of the odds ratio and its interpretation.

Comments: Line 331-3: This sentence isn't very clear - suggest rewording it.

Responses: Thank you for your suggestion. The sentence has been rephrased for clarity as per your suggestions.

Comments: Line 350: compared to the figure for the same city cited in line 98?

Responses: Thank you for your observation. The figure of 0.55 ± 0.17 kg per person per day for Jimma city cited on page 4, line 113 was from a previous study by Getahun et al. (2012), which we used in the introduction to provide context. The current figure of 0.66 kg per person per day reported in our manuscript represents the findings from our study. We have clarified this distinction in the revised manuscript and used 0.66 kg per person per day from our findings.

Comments: Line

---

## [Decision Letter · Decision Letter 1]

12 Aug 2025

PONE-D-25-22742R1Determinants of sustainable solid waste management in Jimma City, Southwest EthiopiaPLOS ONE

Dear Dr. Haile,

Thank you for submitting your manuscript to PLOS ONE. After careful consideration, we feel that it has merit but does not fully meet PLOS ONE’s publication criteria as it currently stands. Therefore, we invite you to submit a revised version of the manuscript that addresses the points raised during the review process.

Reviewer 1 has some further minor comments to address.

We look forward to receiving your revised manuscript.

Kind regards,

Alison Parker

Academic Editor

PLOS ONE

Journal Requirements:

Reviewers' comments:

Reviewer's Responses to Questions

**Comments to the Author**

1. If the authors have adequately addressed your comments raised in a previous round of review and you feel that this manuscript is now acceptable for publication, you may indicate that here to bypass the “Comments to the Author” section, enter your conflict of interest statement in the “Confidential to Editor” section, and submit your "Accept" recommendation.

Reviewer #1: (No Response)

Reviewer #2: All comments have been addressed

2. Is the manuscript technically sound, and do the data support the conclusions?

Reviewer #1: Yes

Reviewer #2: Yes

3. Has the statistical analysis been performed appropriately and rigorously? 

Reviewer #1: Yes

Reviewer #2: Yes

4. Have the authors made all data underlying the findings in their manuscript fully available?

Reviewer #1: Yes

Reviewer #2: Yes

5. Is the manuscript presented in an intelligible fashion and written in standard English?

Reviewer #1: Yes

Reviewer #2: Yes

6. Review Comments to the Author

Reviewer #1: The manuscript has been significantly improved since the first draft - congratulations!

Minor comments:

Lines 90-92: Please state which year(s) these figures refer to. As waste arisings are increasing so quickly, it is useful to know how recent this data is.

Line 105: by “evergreen” do you mean “agricultural”? Replace the full-stop at the end of this short sentence with a comma to incorporate it into the first part of next sentence, making the next sentence shorter.

Line 107: no exception to what? Suggest removing this and starting sentence with “In Jimma City…”

Lines 323 and 333: the numbers are not visible when referring to the Figures.

In table 4, the < symbols are not visible.

The manuscript would benefit from a final proof read for minor grammatical inconsistencies but otherwise reads well.

Reviewer #2: Hello Authors,

Thank you for your careful and thorough revisions. It is clear that you have addressed each of the points raised. The paper now reads with greater clarity and precision, and the changes you have made are evident across several sections.

In the introduction, the correction of referencing inconsistencies, the removal of duplicate entries, and the rectification of the citation swap between references 4 and 5 have strengthened the accuracy of the background. The decision to replace the earlier 30–50% statistic with a more precise figure and source not only improves the reliability of the statement but also brings the framing of the problem into sharper focus.

Your methodology section benefits from a fuller account of how the 9-question scoring tool was developed, combining insights from the literature with input from experts in the field. The explanation for selecting six kebeles, balancing resource limitations with the need for representativeness is now set out with clarity, making the study design easier to follow.

The results section now resolves the earlier uncertainty around household numbers, distinguishing clearly between the 836 initially surveyed and the 820 analysed in detail. Small but important corrections to rounding and typographical error have also been addressed.

In the discussion and conclusion, the replacement of an incorrect reference, the update to the waste generation figure, and the note that references 52 and 53 relate to Bangladesh all add precision. I also welcome the more measured statement regarding the Khat market ban, which now acknowledges the absence of any formal study to support the claim. This adjustment strengthens the balance of the discussion and ensures it rests on secure evidential ground.

These revisions have strengthened the manuscript considerably. It now presents its findings with accuracy, clarity, and a coherent flow. I would recommend it for the next stage in the publication process, and I appreciate the time and care you have invested in addressing the feedback.

All the best!

7. PLOS authors have the option to publish the peer review history of their article (what does this mean? ). If published, this will include your full peer review and any attached files.

**Do you want your identity to be public for this peer review?** For information about this choice, including consent withdrawal, please see our Privacy Policy .

Reviewer #1: **Yes: ** Elizabeth Cullen

Reviewer #2: No

---

## [Author Response · Author response to Decision Letter 2]

27 Aug 2025

Response to editor comments on Journal requirements:

Comment 1: If the reviewer comments include a recommendation to cite specific previously published works, please review and evaluate these publications to determine whether they are relevant and should be cited.

Response: Thank you for your reminder regarding the citation of previously published works. In this case, neither the editor nor the reviewers suggested any additional references to be cited. Therefore, no changes were made to the citations or reference list in the revised manuscript.

Comment 2: Please review your reference list to ensure that it is complete and correct. If you have cited papers that have been retracted, please include the rationale for doing so in the manuscript text, or remove these references and replace them with relevant current references.

Response: Thank you for your critical reminder regarding the reference check. We have carefully reviewed the reference list in the manuscript and confirm that all citations are correct and complete. To the best of our knowledge, none of the cited works has been retracted. Therefore, no changes have been made to the reference list.

---

## [Editor Report · Decision Letter 2]

11 Sep 2025

Determinants of sustainable solid waste management in Jimma City, Southwest Ethiopia

PONE-D-25-22742R2

Dear Dr. Haile,

We’re pleased to inform you that your manuscript has been judged scientifically suitable for publication and will be formally accepted for publication once it meets all outstanding technical requirements.

Kind regards,

Alison Parker

Academic Editor

PLOS ONE
---

## [Editor Report · Acceptance letter]

PONE-D-25-22742R2

PLOS ONE

Dear Dr. Haile,

I'm pleased to inform you that your manuscript has been deemed suitable for publication in PLOS ONE. Congratulations! Your manuscript is now being handed over to our production team.

Kind regards,

on behalf of

Dr. Alison Parker

Academic Editor

PLOS ONE